# Sleep Trajectories in Amnestic and Non-Amnestic MCI: Longitudinal Insights from Subjective and Objective Assessments

**DOI:** 10.3390/diagnostics15212815

**Published:** 2025-11-06

**Authors:** Areti Batzikosta, Despina Moraitou, Paschalis Steiropoulos, Elvira Masoura, Georgia Papantoniou, Ioanna-Giannoula Katsouri, Maria Sofologi, Glykeria Tsentidou, Magda Tsolaki

**Affiliations:** 1Laboratory of Psychology, Department of Cognition, Brain and Behavior, School of Psychology, Faculty of Philosophy, Aristotle University of Thessaloniki (AUTh), 54124 Thessaloniki, Greece; demorait@psy.auth.gr (D.M.); emasoura@psy.auth.gr (E.M.); gltsentidou@gmail.com (G.T.); 2Laboratory of Neurodegenerative Diseases, Center of Interdisciplinary Research and Innovation (CIRI-AUTH), Balcan Center, Buildings A & B, 57001 Thessaloniki, Greece; gpapanto@uoi.gr (G.P.); tsolakim1@gmail.com (M.T.); 3Department of Respiratory Medicine, Medical School, Democritus University of Thrace, 68100 Alexandroupolis, Greece; steiropoulos@yahoo.com; 4Laboratory of Psychology, Department of Early Childhood Education, School of Education, University of Ioannina, 45110 Ioannina, Greece; m.sofologi@uoi.gr; 5Institute of Humanities and Social Sciences, University Research Centre of Ioannina (URCI), 45110 Ioannina, Greece; 6Department of Occupational Therapy, Faculty of Health and Caring Sciences, University of West Attica, 12243 Athens, Greece; ykatsouri@uniwa.gr; 7Department of Psychology, School of Health Sciences, Neapolis University Pafos, 8042 Pafos, Cyprus; 8Greek Association of Alzheimer’s Disease and Related Disorders (GAADRD), Petrou Sindika 13 Str., 54643 Thessaloniki, Greece

**Keywords:** actigraphy, aging, Alzheimer’s disease, mild cognitive impairment, neurodegeneration, sleep disturbances

## Abstract

**Background/Objectives:** Sleep disturbances are increasingly recognized as dynamic biomarkers of cognitive decline; however, longitudinal and multimodal studies directly comparing amnestic (aMCI) and non-amnestic mild cognitive impairment (naMCI) remain limited. **Methods:** In a three-wave longitudinal design (~24 months), 179 older adults (46 healthy controls [HCs], 75 aMCI, 58 naMCI; mean age = 70.2 years, education = 12.3 years) were assessed with actigraphy and validated questionnaires (Athens Insomnia Scale, Pittsburgh Sleep Quality Index, STOP-BANG). Mixed ANOVAs and structural equation modeling tested group, time and mediation effects. **Results:** Subjective measures revealed a progressive worsening of insomnia and sleep quality in MCI, with naMCI exhibiting the steepest decline, while HCs remained largely stable. STOP-BANG trajectories indicated increasing sleep-disordered breathing risk across groups, most pronounced in naMCI. Objective indices corroborated these findings: total sleep time (TST) and sleep efficiency (SE) declined significantly in MCI, especially naMCI, while wake after sleep onset (WASO) increased longitudinally. By the third assessment, naMCI consistently showed the shortest TST and lowest SE. Mediation analyses identified SE as a central predictor of future subjective complaints, with indirect contributions from WASO and PSQI. **Conclusions:** Longitudinal trajectories, rather than cross-sectional comparisons, best differentiated MCI subtypes. NaMCI demonstrated the most aggressive deterioration in both objective and subjective sleep measures, highlighting its heightened vulnerability to sleep dysregulation and potential relevance for neurodegenerative progression. Clinically, sustained monitoring of SE, TST, and sleep-disordered breathing risk may provide prognostic value and inform early, targeted interventions in at-risk populations.

## 1. Introduction

Sleep plays a vital role in maintaining cognitive function, emotional regulation, and overall brain homeostasis. Sleep disturbances have been increasingly linked in accelerated cognitive decline and a heightened risk for neurodegenerative diseases, including Alzheimer’s disease [1,2]. Mild Cognitive Impairment (MCI) represents a transitional state between normal aging and dementia and presents with considerable heterogeneity in its clinical manifestations and prognosis [3]. Recent findings have demonstrated that individuals with MCI frequently experience significant disruptions in sleep. A 2024 meta-analysis reported that subjective sleep disturbances are present in approximately 35.8% of individuals with MCI, while objective assessments reveal disturbances in up to 46.3% [4]. Moreover, a systematic review by Li et al. [5] found that patients with amnestic MCI (aMCI) tend to exhibit reduced sleep efficiency and decreased sleep duration compared to cognitively healthy older adults.

The distinction between amnestic (aMCI) and non-amnestic (naMCI) subtypes of MCI is essential, as they may reflect divergent underlying neuropathological pathways [6]. Diagnostic classification is typically based on the primary cognitive domain affected: aMCI is characterized by predominant episodic memory impairment—either in a single domain or alongside additional deficits—and is more strongly linked to Alzheimer-type pathology [7,8]. In contrast, naMCI involves impairments in non-memory domains such as attention, executive function, visuospatial abilities, or language, while memory remains relatively preserved. This subtype has been associated with alternative etiologies, including vascular factors, Lewy body pathology, or frontotemporal degeneration [9]. However, the differential impact of sleep disturbances across these subtypes remains understudied. Some evidence suggests that aMCI is more closely associated with sleep architecture changes relevant to memory systems, including hippocampal-dependent NREM alterations [10]. Conversely, naMCI may involve distinct sleep–wake disruptions potentially related to deficits in executive or attentional processes [11].

Beyond behavioral differences, emerging evidence indicates that sleep disturbances may contribute to cognitive decline, potentially through mechanisms involving neurodegeneration [12]. Neuroimaging and cerebrospinal fluid biomarkers, such as β-amyloid and tau levels, have been associated with disrupted sleep architecture and impaired slow-wave sleep, which may affect glymphatic clearance and cognitive functioning [13]. Although our study does not include direct biomarker or neuroimaging data, examining longitudinal sleep and cognition in this well-characterized cohort provides valuable behavioral insights and complements ongoing research in this area.

Despite growing recognition of the bidirectional relationship between sleep and cognitive impairment, longitudinal studies examining both objective and subjective sleep across MCI subtypes remain scarce [14]. Existing research is often constrained by cross-sectional designs and reliance on single measurement modalities, limiting the ability to identify dynamic patterns of sleep-related cognitive risk.

The present study addresses this gap by examining sleep in three groups, cognitively healthy older adults, individuals with aMCI, and individuals with naMCI, across three time points. Objective sleep parameters were measured using wrist actigraphy, and subjective sleep quality was assessed through standardized self-report questionnaires. By integrating multimodal sleep data within a longitudinal framework, the present study aims to identify distinct sleep trajectories that clearly differentiate MCI subtypes and may serve as early behavioral markers of cognitive decline.

### Hypotheses of the Present Study

Taking into consideration the theoretical background and neurodegenerative models of cognitive decline, the following hypotheses were formulated:Both MCI groups were expected to exhibit significantly poorer subjective and objective sleep quality than healthy controls, with differences clearer across longitudinal trajectories and to be more pronounced in naMCI.Sleep quality was expected to decline over time, most markedly in naMCI, then aMCI; healthy controls will remain relatively stable or show only minimal age-related changes.Significant group-by-time interaction effects were anticipated: MCI participants will exhibit steeper deterioration in sleep parameters over time than controls.Early objective sleep metrics (total sleep time, sleep efficiency, and wake after sleep onset) were expected to significantly predict subjective sleep quality changes (as measured by AIS, PSQI, and STOP-BANG) at the later time points.

## 2. Materials and Methods

### 2.1. Design

This longitudinal study comprised three assessment waves over approximately 2 years. Following the baseline evaluation, participants completed two additional assessments at comparable intervals. All waves adhered to an identical standardized protocol, including both objective and subjective sleep measures.

### 2.2. Participants

An a priori power analysis using G*Power version 3.1.9.7 indicated that a minimum of 148 participants was required to achieve 80% statistical power [15]. Based on this, 185 participants were initially recruited and completed comprehensive cognitive assessments. Prior to the main analyses, a predefined screening procedure led to the exclusion of six participants to ensure sample homogeneity and reduce potential confounding factors: one individual presented with Subjective Cognitive Impairment (SCI) characterized by persistent self-reported cognitive complaints, one had a confirmed diagnosis of early dementia, and four were undergoing pharmacological treatment for sleep disorders. These exclusions were determined by the principal investigator in collaboration with the research team, and the data from these participants were not included in subsequent analyses, yielding a final sample of 179 participants.

The final cohort included 179 participants, categorized into three groups: 46 cognitively healthy controls, 75 individuals diagnosed with aMCI, and 58 with naMCI. The sample comprised 53 men and 126 women, with a mean age of 70.23 years (SD = 4.74) and an average educational level of 12.35 years (SD = 3.22) (Table 1).

Throughout the study period, diagnostic stability was confirmed across all assessment points; no cases of transition between naMCI and aMCI or progression to dementia were identified.

### 2.3. Inclusion Criteria

Eligibility criteria required participants to be aged 65 years or older and to have completed at least 6 years of formal education. Based on a comprehensive neuropsychological assessment, conducted by the Greek Association of Alzheimer’s Disease, participants were classified into three groups according to their cognitive status: cognitively healthy controls, individuals with aMCI, or those with naMCI. The assessment was conducted in accordance with Petersen’s diagnostic criteria [16] and the Diagnostic and Statistical Manual of Mental Disorders, Fifth Edition, Text Revision (DSM-5-TR) [17], and was accompanied by clinical and biological evidence, including MRI findings and cerebrospinal fluid (CSF) biomarkers indicative of neurodegenerative processes. However, we did not have access to the participants’ neuroimaging or CSF data, as these assessments were conducted by the Greek Association of Alzheimer’s Disease as part of their diagnostic protocol prior to sample provision.

The control group included 46 cognitively healthy adults residing in the community. All participants exhibited intact cognitive functioning, as indicated by their Montreal Cognitive Assessment (MoCA) scores, which ranged from 27 to 30 (M = 27.8, SD = 0.74), confirming their classification as cognitively normal.

The aMCI group comprised 75 individuals who had received a diagnosis of MCI within the preceding 2 years. Diagnoses were established in accordance with DSM-5-TR criteria for Mild Neurocognitive Disorders [17] and further corroborated through comprehensive neuropsychological assessments, neurological examinations, neuroimaging, and psychiatric evaluations. Inclusion required (a) a confirmed diagnosis of Minor Neurocognitive Disorder and (b) a score at least 1.5 standard deviations (SD) below the normative mean on an episodic memory task. All participants in the aMCI group (*n* = 75) were classified as having the amnestic subtype, with MoCA scores ranging from 24 to 29 (M = 24.59, SD = 2.7).

The naMCI group included 58 individuals who had been diagnosed with MCI within the previous 2 years. Similarly to the aMCI group, inclusion criteria required (a) a confirmed diagnosis of Minor Neurocognitive Disorder and (b) a score at least 1.5 standard deviations (SDs) below the normative mean in at least one cognitive domain other than memory, based on the results of comprehensive neuropsychological evaluations. Following evaluation, all 58 participants were classified as having the non-amnestic subtype of MCI, with MoCA scores ranging from 20 to 29 (M = 26.10, SD = 2.47).

### 2.4. Exclusion Criteria

Participants were excluded if they had (a) a history of psychiatric disorders, (b) substance use or alcohol dependence, (c) prior traumatic brain injury, (d) neurological conditions, (e) diagnosed sleep disorders, (f) current use of medications for sleep-related issues, (g) use of antidepressants or anxiolytics, or (h) cognitive complaints and/or a diagnosis of MCI or any form of dementia in the healthy control group.

All participants underwent a comprehensive neuropsychological evaluation at the Greek Association of Alzheimer’s Disease and Related Disorders to determine their diagnostic status. Mood and affective disorders were assessed using the Geriatric Depression Scale [18,19], Beck Depression Inventory [20], Beck Anxiety Inventory [21], and Short Anxiety Screening [22,23], while the Neuropsychiatric Inventory [24,25] was used to detect neuropsychiatric symptoms.

Cognitive functioning was evaluated using the Mini-Mental State Examination (MMSE) [26,27] and the Montreal Cognitive Assessment (MoCA) [28,29], providing an overall measure of cognition. Executive functions were further assessed through the Functional Cognitive Assessment [30], which includes tasks simulating daily living activities, alongside a battery of standardized tests covering memory, attention, executive function, and language. The Global Deterioration Scale (GDS) [31] was applied to rate the severity of cognitive decline, with stage 1 representing no cognitive decline and stage 3 corresponding to MCI. For a full description of the neuropsychological tests, see Tsolaki et al. [32].

Statistical analyses indicated no significant differences between the three groups in terms of age, F (2, 176) = 2.977, *p* = 0.054, years of education, F (2, 176) = 0.452, *p* = 0.637, or gender distribution, χ^2^ = 1.849, *p* = 0.397.

### 2.5. Ethics

All participants were thoroughly informed about the study’s objectives through both verbal explanations and written documentation, with explicit assurances regarding the confidentiality of their personal information. Written informed consent was obtained prior to baseline assessment, confirming voluntary participation and clarifying that participants could withdraw at any time without any consequences. Demographic information, including age, gender, and educational attainment, was collected in accordance with European Union regulations under the General Data Protection Regulation (effective 28 May 2018), which permits the use of sensitive personal data for research purposes. Participants were also informed of their right to request the deletion of their data from the study database. The study protocol received approval from the Scientific and Ethics Committee of the Greek Association of Alzheimer’s Disease and Related Disorders (Approval Code: 29/15-02-2017) and adhered to the ethical principles outlined in the Declaration of Helsinki.

### 2.6. Procedure

Participant recruitment was facilitated by undergraduate psychology interns at the Daycare Centers of the Greek Association of Alzheimer’s Disease and Related Disorders. Eligible individuals were invited to participate, and those who agreed were contacted by the study psychologist. The psychologist provided detailed information about the study’s purpose and procedures. Following this, participants scheduled two morning appointments within 1 week, each lasting up to 1 h, to complete the necessary assessments. Informed written consent was obtained during the first appointment, ensuring participants’ confidentiality and explaining the objectives of the study.

Given the longitudinal nature of the research, participants were contacted for follow-up assessments at two additional time points: the second assessment (about 8 months after the initial evaluation) and the third assessment (about 8 months following the second). Each follow-up phase adhered to the same assessment procedures as the initial phase to ensure consistency across time points. The assessments were conducted individually in a quiet, comfortable environment, using two different versions of the testing battery to minimize order effects. Participants were not compensated for their involvement in the study.

### 2.7. Instruments

#### 2.7.1. Objective Assessment of Sleep

##### Actigraphy

Actigraphy is a technique employed to assess sleep–wake patterns by analyzing movement data [33]. It is widely used in research due to its ability to monitor activity in naturalistic, real-world settings. In this study, all participants wore an actigraphy device for a duration of 7 days. The wrist-worn device, similar in appearance to a watch, records movement, and this data point is subsequently processed to provide insights into the individual’s sleep–wake cycle. The Philips Respironics Actiwatch Spectrum Pro (version 5.57.0006) was used for monitoring purposes. The device records both motion and light exposure, providing essential information on activity levels, sleep patterns, and overall sleep quality. Actigraphy relies on accelerometer-based motion detection rather than eye movement data and provides indirect but reliable estimates of sleep–wake behavior. It is commonly applied in sleep medicine to assess sleep disorders, circadian rhythm disturbances, and daily activity [34]. Parameters such as sleep onset, total sleep duration (TST), wake after sleep onset (WASO), and sleep efficiency (SE) are automatically calculated and presented via a basic reader, enabling data visualization on a computer or tablet [35].

#### 2.7.2. Subjective Measures of Sleep

Athens Insomnia Scale: The scale (AIS) is an eight-item self-assessment questionnaire used to evaluate insomnia severity in adults, covering aspects like sleep onset, maintenance, and daytime consequences. Each item is rated on a scale from 0 to 3, with higher scores indicating more severe insomnia. The AIS has been validated for use in the Greek population, showing high sensitivity (93%) and specificity (85%) for diagnosing insomnia, with a total score of 6 or higher as the optimal cutoff for identifying cases [36].

STOP-BANG Questionnaire: The questionnaire is an eight-question screening tool used to evaluate the risk of obstructive sleep apnea (OSA) [37]. It assesses factors such as snoring, tiredness, observed apneas, high blood pressure, body mass index (BMI), age, neck circumference, and gender. Each “yes” response scores one point, with a total possible score of 8, where higher scores indicate a greater likelihood of OSA. The tool has been validated in the Greek population, demonstrating that a score of 4 or more effectively identifies individuals with OSA [38].

The Pittsburgh Sleep Quality Index (PSQI): It is a 19-item self-report tool designed to evaluate sleep quality over the past month, encompassing seven areas including sleep latency, duration, and disturbances [39]. Scores range from 0 to 21, with higher scores reflecting worse sleep quality. The Greek version of the PSQI has been validated and is considered highly reliable and effective for both clinical and research purposes, including among individuals with sleep disorders and cancer patients [40,41,42].

Although qualitative sleep diaries were initially included to complement subjective reporting, they were later excluded due to inconsistent completion among participants.

### 2.8. Statistical Analysis

All analyses were performed in IBM SPSS Statistics version 29.0 [43] and JASP version 0.18.3 [44]. Mixed-design ANOVAs were first conducted to examine both between- and within-subject effects, with Group (HCs, aMCI, naMCI) as the between-subject factor and Time (baseline, 2nd, 3rd assessment) as the within-subject factor. When significant main or interaction effects emerged, separate repeated-measures ANOVAs (for within-group comparisons across time) or one-way ANOVAs (for between-group comparisons at each time point) were performed to further explore the effects. Multivariate tests in the mixed-design ANOVAs were reported using Pillai’s Trace, a robust statistic assessing the overall effect of independent variables on a set of correlated dependent variables. Standard assumption testing was applied (sphericity, homogeneity of variance/covariance). Effect sizes are reported as partial η^2^, and Scheffé post hoc tests were conducted for multiple comparisons. Mediation analyses were conducted using structural equation modeling (SEM) to examine the dynamic relationships between objective and subjective sleep parameters across time. Objective sleep measures (TST, SE, WASO) at the second assessment were used to predict changes in subjective sleep outcomes (AIS, PSQI, STOP-BANG) at the third assessment, with mediators including subjective sleep measures at the second assessment and objective measures at the third. This approach allowed us to capture both cross-lagged effects—how earlier measures influence subsequent ones—and autoregressive effects—how each variable predicts its own future values. Analyses were conducted on the total sample due to sample size requirements, which did not allow SEM to be run separately for each group.

## 3. Results

Among the objective sleep parameters recorded via actigraphy, only those that showed statistically significant findings are included in the results presented below, to maintain clarity and relevance in the findings.

### 3.1. Objective Sleep Measures

Total Sleep Time: A 3 × 3 mixed-design ANOVA was conducted with diagnostic group (HCs, aMCI, naMCI) as the between-subjects factor and time (three assessment points) as the within-subjects factor. There was a significant main effect of group, F (2, 176) = 17.47, *p* < 0.001, η^2^ = 0.166, indicating overall differences in total sleep time across groups. A significant main effect of time also emerged, F (1.524, 268.249) = 26.14, *p* < 0.001, η^2^ = 0.129, suggesting changes in TST over time. Crucially, a significant group × time interaction was found, F (3.048, 268.249) = 30.19, *p* < 0.001, η^2^ = 0.255, indicating that changes in TST over time differed by group.

A subsequent MANOVA confirmed a significant multivariate effect of diagnostic group on TST across time points, Pillai’s Trace, V = 0.428, F (6, 350) = 15.89, *p* < 0.001, η^2^ = 0.214. Follow-up univariate tests revealed significant group differences at all three time points. Post hoc Scheffe tests showed that the naMCI group had significantly shorter TST than healthy controls at both the second, I-J = −83.14, *p* < 0.001 and third, I-J = −82.40, *p* < 0.001 assessment.

Regarding the effect of time of assessment on TST, repeated measures ANOVA was conducted separately for each group. The analyses revealed statistically significant changes in TST scores across the three time points, F (2.44) = 17.901, *p* < 0.001, η^2^ = 0.44. Among HCs, pairwise comparisons indicated a statistically significant increase in TST between the first and second assessments, I-J = −37.348, *p* < 0.001, while no significant differences were observed between other time points. These results suggest a modest but significant fluctuation in sleep duration over time within the HC group.

Within the aMCI group, repeated measures ANOVA demonstrated a significant variation in TST across the three assessment sessions, F (2.73) = 24.887, *p* < 0.001, η^2^ = 0.40. Pairwise comparisons revealed a significant reduction in TST between the first and third evaluations, I-J = −7.680, *p* = 0.001, as well as between the second and third time points, I-J = −7.453, *p* = 0.001. This pattern suggests a notable and continuous decrease in sleep duration across the measured time points in the aMCI group.

In the naMCI group, a significant and progressive decline in TST was observed across the three assessment points, F (2.56) = 40.592, *p* < 0.001, η^2^ = 0.59. Pairwise comparisons revealed a significant reduction in TST scores from the first to the third assessment, I-J = −13.397, *p* = 0.001. Sleep duration also declined significantly between the second and the third assessment, I-J = −11.172, *p* = 0.001. These findings indicate a substantial deterioration in sleep duration over time among individuals with naMCI (Figure 1).

Sleep Efficiency: A 3 × 3 mixed-design ANOVA revealed no significant main effect of diagnostic group on sleep efficiency. However, a significant main effect of time was observed, F (1.385, 243.763) = 37.82, *p* < 0.001, η^2^ = 0.177, indicating that SE changed over time. Importantly, a significant group × time interaction was found, F (2.770, 243.763) = 37.03, *p* < 0.001, η^2^ = 0.296, suggesting that the pattern of SE change over time differed between groups.

A subsequent MANOVA revealed a significant multivariate effect of diagnostic group on SE across time points, Pillai’s Trace V= 0.379, F (6, 350) = 13.64, *p* < 0.001, η^2^ = 0.190. Follow-up univariate tests indicated no significant group differences at the first assessment, but significant differences emerged at the second, F (2, 176) = 3.99, *p* = 0.020, η^2^ = 0.043, and third assessments, F (2, 176) = 6.15, *p* = 0.003, η^2^ = 0.065. Post hoc Scheffe comparisons showed that participants with naMCI had significantly lower SE than healthy controls at both the second, I-J = −4.99, *p* = 0.022 and third assessment, I-J = −6.15, *p* = 0.003.

The three groups exhibited statistically significant differences in their sleep efficiency scores across the three assessment times. Specifically, in HCs, repeated-measures analysis revealed a significant decline in SE over time, F (2.444) = 16.756, *p <* 0.001, η^2^ = 0.43. Pairwise comparisons indicated that the mean SE score significantly decreased only between the first and second assessment, I-J = −3.283, *p* = 0.001.

Analysis of SE scores within the aMCI group across three time points revealed a significant deterioration in sleep efficiency over time, F (2, 73) = 95.380, *p* < 0.001, η^2^ = 0.72. Pairwise comparisons indicated a progressive decrease in SE scores from the first to the third assessment (I-J = −1.987, *p* = 0.001), as well as a significant difference between the second and third assessments (I-J = −1.707, *p* = 0.001).

Analysis of SE scores within the naMCI group revealed a significant and progressive deterioration in sleep efficiency over time, F (2, 56) = 53.313, *p* < 0.001, η^2^ = 0.65. Pairwise comparisons indicated a significant decrease in SE scores from the first to the second assessment, I-J = −0.845, *p* = 0.001, with a further significant increase from the first to the third assessment, I-J = −3.000, *p* = 0.001. The difference between the second and the third assessment was also significant, I-J = −2.155, *p* = 0.001. These results suggest a marked and progressive decline in sleep efficiency over time in individuals with naMCI (Figure 2).

Wake After Sleep Onset: A 3 × 3 mixed-design ANOVA revealed no significant main effect of diagnostic group on WASO. A significant main effect of time emerged, F (1.858, 326.952) = 39.03, *p* < 0.001, η^2^ = 0.182, suggesting changes in WASO over time. Crucially, a significant group × time interaction was found, F (3.715, 326.952) = 12.11, *p* < 0.001, η^2^ = 0.121, indicating that changes in WASO over time differed by group.

Although no significant between-group differences were detected at any individual time point, within-group longitudinal effects emerged. Specifically, the aMCI group demonstrated statistically significant differences in their WASO scores across assessment time points. The aMCI group exhibited a progressive worsening of WASO, F. (2.73) = 39.162, *p* < 0.001, η^2^ = 0.51, with significant increases from the first to the third assessment, I-J = −6.360, *p* = 0.001, as well as between the second and third assessment, I-J = −5147, *p* = 0.001.

In the naMCI group, a significant and progressive increase in WASO scores was observed over time, F (2.56) = 37.712, *p* < 0.001, η^2^ = 0.57. Pairwise comparisons revealed a significant increase in WASO scores from the first time of assessment to the second, I-J = 2.862, *p* = 0.001. WASO also increased significantly between the first and the third assessment, I-J = 8.828, *p* = 0.001. Additionally, a significant increase was noted between the second and third assessments, I-J = −5.966, *p* = 0.001 (Figure 3).

### 3.2. Subjective Sleep Assessment

Athens Insomnia Scale: Initially, a 3 × 3 mixed-design ANOVA was conducted with group (HCs, aMCI, and naMCI) as the between-subjects factor and time of assessment (three time points) as the within-subjects factor. The main effect of diagnostic group was not significant, whereas the main effect of time of assessment was significant, F (1, 176) = 141.73, *p* < 0.001, η^2^ = 0.446, indicating a strong linear increase over time. Furthermore, a significant group × time interaction was observed, F (2, 176) = 20.04, *p* < 0.001, η^2^ = 0.185. Subsequently, analysis of variance was performed with group as the independent variable and performance at each of the three assessment times as dependent variable. According to Pillai’s Trace, group effect was significant, V = 0.229, F (6, 350) = 7.551, *p* < 0.001, η^2^ = 0.115. The difference was only at the first assessment: F (2, 176) = 2.69, *p* = 0.070, η^2^ = 0.030, indicating that group differences in AIS scores were mainly present at baseline and tended to diminish over time. Repeated measures ANOVA was conducted for every group separately, to further examine the effects of time of assessment. For healthy controls, no significant results emerged. However, for the aMCI group, performance gradually worsened: F (2.73) = 36.785, *p* < 0.001, η^2^ = 0.50. Scheffe post hoc comparisons showed a significant increase in insomnia severity from the first to the second assessment, I-J = −0.267, *p* = 0.001, as well as from the first to the third assessment, I-J = −0.733, *p* = 0.001. Furthermore, a significant difference was also observed between the second and the third assessment with higher AIS scores in the third phase, I-J = −0.467, *p* = 0.001. For the naMCI group, performance also worsened over time: F (2.56) = 40.612, *p* < 0.001, η^2^ = 0.59. Significant differences were found between the first and second assessments, I-J = −0586, *p*= 0.001, the first and third assessments, I-J = −1.259, *p* = 0.001, and the second and third assessments, I-J = −0.672, *p* = 0.001. These results indicate a gradual and statistically significant worsening of insomnia symptoms over time (Figure 4).

Pittsburgh Sleep Quality Index: A 3 × 3 mixed-design ANOVA was conducted with group (HCs, aMCI & naMCI) as the between-subjects factor and time of assessment (three time points) as the within-subjects factor. The main effect of diagnostic group was not significant. However, the main effect of time of assessment was significant, F (1.68, 294) = 129.20, *p* < 0.001, η^2^ = 0.425, indicating a pronounced change over time. Moreover, a significant group × time interaction was observed, F (3.36, 294) = 6.34, *p* < 0.001, η^2^ = 0.068, suggesting that the pattern of change over time differed across diagnostic groups. Subsequently, analysis of variance was performed with group (HCs, aMCI, and naMCI) as the independent variable and PSQI scores at each of the three assessment times as dependent variable. A significant effect of diagnostic group was found according to Pillai’s Trace, V = 0.140, F (6, 348) = 4.35, *p* < 0.001, η^2^ = 0.070. One-way ANOVA revealed no significant group effects, indicating that diagnostic groups did not differ significantly in PSQI scores at any assessment.

Additionally, repeated measures ANOVAs were applied separately to the data of each group. For the HCs, repeated-measures ANOVA revealed a significant deterioration in sleep quality over time, F (2.176) = 90.510, *p <* 0.001, η^2^ = 0.50. Pairwise comparisons indicated that PSQI scores significantly increased from the first to the second assessment, I-J = −0.283, *p* = 0.001, and from the first to the third assessment, I-J = −0.435, *p* = 0.001. A significant difference was also found between the second and the third assessment, I-J = −0.152, *p* = 0.001. These findings indicate a progressive and statistically significant decline in sleep quality over time in the healthy control group.

Within the aMCI group, repeated measures ANOVA revealed a significant deterioration in sleep quality over time as well, F (2.72) = 29.281, *p <* 0.001, η^2^ = 0.44. Pairwise comparisons showed a progressive increase in PSQI scores from the first to the second assessment, I-J = −0.284, *p* = 0.001, and a more pronounced increase from the first to the third assessment, I-J = −0.689, *p* = 0.001. A significant difference was also observed between the second and third assessment, I-J= −0.405, *p* = 0.001. These findings suggest a gradual and statistically robust decline in sleep quality among individuals with aMCI.

Within the naMCI group, repeated measures ANOVA revealed a significant and progressive decline in sleep quality over time, F (2.56) = 64.275, *p <* 0.001, η^2^ = 0.69. Pairwise comparisons indicated a significant increase in PSQI scores from the first to the second assessment, I-J = −0.397, *p* = 0.001, and a more pronounced increase from the first to the third assessment, I-J = −0.966, *p* = 0.001. Sleep quality also declined significantly between the second and the third assessment, I-J = −0.569, *p* = 0.001. These results suggest a marked and consistent worsening of sleep quality in individuals with naMCI, possibly even more pronounced than in those with aMCI (Figure 5).

STOP-BANG: Again, a 3 × 3 mixed-design ANOVA was conducted with group (HCs, aMCI, and naMCI) as the between-subjects factor and time of assessment (three time points) as the within-subjects factor. The main effect of diagnostic group was not significant. The main effect of time of assessment was significant, F (1.817, 319.801) = 104.15, *p* < 0.001, η^2^ = 0.372, indicating a pronounced change over time. Furthermore, a significant group × time interaction was observed, F (3.634, 319.801) = 12.92, *p* < 0.001, η^2^ = 0.128, suggesting that the trajectory of change over time varied between diagnostic groups. Subsequently, analysis of variance was performed with diagnostic group (HCs, aMCI, and naMCI) as the independent variable and PSQI scores at each of the three assessment times as dependent variable. The results indicated a significant effect of group, as evidenced by Pillai’s Trace, V = 0.211, F (6, 350) = 6.89, *p* < 0.001, η^2^ = 0.106. No significant group differences in effect were observed at any time point, suggesting that diagnostic groups did not differ significantly in STOP-BANG scores at any assessment.

At the next step, repeated measures ANOVAs were conducted for each group. In the HC group, a statistically significant main effect of time was observed: F (2.44) = 8.667, *p <* 0.001, η^2^ = 0.28. Pairwise comparisons revealed a significant increase in STOP-BANG scores only between the first and third assessments, I-J = −0.283, *p* < 0.001, indicating deterioration in sleep-related breathing patterns over time among cognitively healthy individuals.

Within the aMCI group, repeated measures ANOVA revealed a statistically significant effect of time on STOP-BANG scores, indicating a progressive increase in sleep-related respiratory disorders, F (2.73) = 20.487, *p <* 0.001, η^2^ = 0.36. Pairwise comparisons showed significant increase in STOP-BANG scores from the first to the second assessment, I-J = −0.240, *p* = 0.001, as well as a more pronounced increase from the first to the third assessment, I-J = −0.427, *p* = 0.001. A significant difference was also observed between the second and third assessments, I-J = −0.187, *p* = 0.001, supporting the presence of a consistent deterioration over time in this clinical group.

In the naMCI group, repeated measures ANOVA demonstrated a statistically significant and progressive increase in STOP-BANG scores over time, indicating a heightened risk for sleep-disordered breathing, F (2.56) = 62.612, *p <* 0.001, η^2^ = 0.69. Pairwise comparisons revealed significant increase in STOP-BANG scores between the first and the second assessment, I-J = −0.310, *p* = 0.001, as well as a more pronounced increase between the first and third assessments, I-J = −0.879, *p* = 0.001. A statistically significant difference was also observed between the second and third assessments, I-J = −0.569, *p* = 0.001. These findings suggest a substantial and progressive deterioration in sleep-related respiratory function over time in individuals with naMCI, potentially exceeding that observed in the aMCI group (Figure 6).

### 3.3. Mediation Analysis

A series of different mediation models were tested to reveal the dynamic role of objective assessments in the formulation and change of subjective assessments of sleep, longitudinally, in the total sample (this decision was taken due to N prerequisites for running such an analysis in every group, separately). In the confirmed model, the objective sleep parameters TST, WASO, and SE at second assessment were entered as predictor variables. The outcome variables were the subjective estimations at the third assessment, namely the AIS, the PSQI, and the STOP-BANG questionnaire scores. The mediators were the same subjective sleep parameters as measured at the second time of assessment as well as the objective measures as their scores formulated at the third time of assessment (see Table 2). Data from the first assessment were not included in the mediation model, as the effects were not statistically significant; these results are reported in our previous publication [11].

As shown in Table 2, SE at the second assessment significantly predicted insomnia complaints at the third, whereas TST and WASO showed no direct effects. Indirect pathways were observed, with TST and SE exerting effects on later AIS via their stability across waves, and both WASO and SE influencing PSQI trajectories through prior PSQI scores. Overall, SE emerged as the most consistent longitudinal predictor.

## 4. Discussion

Building on the present findings, the current discussion integrates the longitudinal patterns of objective and subjective sleep disturbances observed in older adults with MCI and healthy controls, offering an interpretation of the mechanisms underlying differential trajectories and their implications for cognitive aging.

### 4.1. Objective Sleep Measures

Objective sleep data obtained via actigraphy revealed complex group-specific patterns and longitudinal changes. Consistent with our hypotheses, TST and SE progressively declined over time in participants with MCI, with the greatest deterioration observed in naMCI. WASO increased primarily in the MCI groups, confirming the hypothesis of longitudinal worsening of objective sleep parameters in clinical populations [45,46]. Regarding TST, our findings are consistent with previous research linking reduced total sleep time to an elevated risk of cognitive decline and faster progression to dementia [47]. While prior studies suggest that mechanisms such as impaired slow-wave sleep and reduced glymphatic clearance of β-amyloid and tau may contribute to this association [48,49,50,51], our study did not directly assess these processes, and therefore we cannot draw conclusions regarding the underlying mechanisms.

The progressive TST decline in naMCI suggests that sleep duration disruptions may reflect early changes related to cognitive decline, particularly in executive and non-memory functions [52]. However, sleep disruptions alone cannot be considered definitive biomarkers of neurodegeneration. Instead, they should be interpreted as complementary to other established biochemical or neuroimaging markers.

Interestingly, HCs showed a transient increase in TST, likely reflecting normal age-related adjustments or circadian-related fluctuations [46]. Although SE differences were absent at baseline, the subsequent progressive decline in SE among naMCI and aMCI highlights the dynamic nature of sleep disturbances [53]. Significant SE reductions at later assessments are consistent with the literature linking low SE to increased amyloid and tau accumulation, as well as hippocampal and prefrontal atrophy [54]. The steeper SE decline in naMCI supports the hypothesis that reduced sleep efficiency is more closely related to executive dysfunction and fronto-striatal network pathology, in contrast to the more memory-specific deficits in aMCI [11,55].

Longitudinal increases in WASO in MCI indicate progressive disruption of sleep architecture. Frequent nighttime awakenings have been associated with glymphatic dysfunction and impaired clearance of neurotoxic metabolites, potentially accelerating cognitive decline [56,57]. The more pronounced and progressive WASO increase in naMCI, compared with aMCI, aligns with evidence that continuity disruptions are more tightly linked to non-memory cognitive pathways and prefrontal network involvement [58,59].

Overall, these findings demonstrate that objective sleep measures differentiate MCI subgroups longitudinally, with naMCI exhibiting the most pronounced and progressive impairments. While reductions in TST and SE were most diagnostic, increases in WASO were primarily informative longitudinally, supporting the use of actigraphy as a sensitive prognostic tool for MCI progression. By contrast, healthy controls maintained relatively stable WASO trajectories, with only minor age-related fluctuations, clearly distinguishing them from both MCI groups. This separation underscores the pathological specificity of WASO increases in MCI and strengthens the case for actigraphy-derived indices as sensitive biomarkers [60]. Overall, these findings demonstrate that objective sleep measures not only differentiate MCI subgroups longitudinally, with naMCI exhibiting the steepest deterioration, but also distinguish clinical groups from healthy aging, supporting their use as prognostic tools for cognitive decline [61].

### 4.2. Subjective Sleep Measures

The trajectories of subjective sleep estimates showed that insomnia symptoms worsened significantly over time in both aMCI and naMCI, with the largest effect observed in naMCI, whereas healthy controls remained largely stable. The pronounced group × time interaction indicates that insomnia symptomatology may become increasingly evident as neural networks regulating sleep–wake transitions and sleep maintenance, such as the prefrontal–thalamo–cortical loop, progressively decline. Although no consistent cross-sectional differences were observed in our sample [11], these longitudinal trajectories are consistent with previous literature linking poorer subjective sleep quality to accelerated cognitive decline and, in other studies, to neuroaxonal damage biomarkers such as elevated plasma neurofilament light chain (pNFL) [62].

Progressive deterioration in overall sleep quality, as measured by the PSQI, was evident across all groups, with the most pronounced increase in naMCI. This pattern likely reflects heightened vulnerability of executive and attentional networks in this subgroup, compounded by vascular comorbidities, circadian rhythm instability, and age-related environmental factors (e.g., reduced melatonin, polypharmacy, lifestyle changes). The steeper PSQI slope in naMCI underscores the interaction between these factors and early neurodegenerative processes. Consistent with prior work, poorer PSQI scores are associated with increased likelihood of MCI and faster cognitive decline [62,63].

Risk for sleep-disordered breathing (SDB), assessed via STOP-BANG, increased over time in all groups, again with the steepest rise in naMCI. This trajectory aligns with evidence linking obstructive sleep apnea (OSA) risk to early executive and attentional deficits and accelerated cognitive decline [64]. Even modest increases in STOP-BANG scores may shift participants above clinical thresholds (≥4), highlighting the potential clinical significance of longitudinal monitoring. Pre-symptomatic screening for OSA in MCI populations is therefore recommended, due to frequent underdiagnosis [65].

Across all subjective measures, longitudinal analyses provided more discriminative and prognostic information than cross-sectional comparisons, highlighting the dynamic nature of sleep changes in aging and early cognitive decline [52,56,57,60]. While Hypothesis 1, predicting baseline group differences, was not fully supported, partial confirmation of Hypothesis 2 was observed: AIS and PSQI scores progressively worsened in both aMCI and naMCI groups over time, whereas HCs remained relatively stable, indicating that sleep deterioration emerges gradually rather than at a single time point [46,62,63]. For STOP-BANG, progression followed the expected hierarchy (naMCI > aMCI > HCs), reflecting a cumulative increase in risk for obstructive sleep apnea (OSA) across cognitive subgroups [64,65,66,67]. Hypothesis 3, predicting significant group × time interactions, was confirmed, with naMCI consistently exhibiting the steepest slopes, particularly for global sleep quality (PSQI) and OSA risk (STOP-BANG) [68,69,70]. These patterns suggest that sleep disruption in MCI is multifaceted, encompassing insomnia, overall sleep quality, and susceptibility to SDB, with longitudinal trajectories providing critical insights into disease-related vulnerability beyond what cross-sectional analyses can reveal.

Neurologically, changes in PSQI and AIS may reflect medial temporal and hippocampal alterations affecting memory-dependent sleep processes, whereas STOP-BANG progression implicates fronto-subcortical and vascular pathways associated with apnea and hypoxia [71,72]. The sharper deterioration in naMCI likely reflects increased susceptibility of executive and attentional networks to sleep dysregulation, distinguishing it from aMCI. Stability in HCs reinforces that these trajectories are primarily disease-related rather than normative aging phenomena [73]. The observed divergence between subjective and objective sleep measures, particularly in pathological aging, may explain the absence of consistent cross-sectional differences [52,74].

These findings underscore that slope analyses are more diagnostically and prognostically informative than cross-sectional comparisons, supporting contemporary models showing divergent sleep trajectories in aging and preclinical dementia, some detectable first in subjective measures [64]. The well-documented discrepancy between subjective and objective sleep especially in pathological aging likely contributed to the absence of consistent cross-sectional differences and must be considered in interpretation [52,74].

In sum, despite the absence of consistent point-in-time group differences, longitudinal trajectories of subjective sleep measures reveal a clear pattern of worsening in MCI, with naMCI showing the fastest decline. These findings align with recent evidence identifying sleep as a dynamic behavioral biomarker in preclinical and early cognitive decline and support regular longitudinal monitoring and proactive pre-symptomatic SDB assessment in general and clinical populations [62,65,70].

### 4.3. The Interplay Between Objective Sleep Parameters and Longitudinal Changes in Subjective Sleep Perceptions

The mediation model revealed complex effects of objective scores on subjective sleep indices. Specifically, SE at the second assessment was the only index that directly predicted AIS scores at the last assessment, indicating that perceived insomnia is influenced by objective sleep quality, as this index is formulated longitudinally. These findings highlight the importance of monitoring SE over time and indicate that improvements in this index may have a meaningful impact on reducing perceived insomnia, whereas other objective indices appear to have less predictive value in the long term [75].

Indirect effects revealed in the longitudinal model point to nuanced dynamics. Sleep efficiency at the second assessment predicted subsequent AIS scores not only by its instantaneous level, but also via its trajectory, such that sustained efficiency buffered against worsening perceived insomnia, whereas progressive declines exacerbated it. These findings suggest that objective sleep indices shape long-term subjective perceptions in MCI. While subjective sleep complaints may not always faithfully mirror physiological sleep, they appear to capture meaningful downstream consequences of objective alterations, supporting their clinical relevance. Indeed, sleep misperception [76], the common mismatch between subjective report and objective measures, is well-documented in MCI populations, where objective–subjective correlations are often weak, highlighting that subjective measures likely reflect memory-related recall bias or altered self-awareness rather than precise physiologic changes [77].

Clinically, these results indicate that longitudinal monitoring of SE and WASO can provide reliable indicators of future subjective sleep disturbances in MCI. Recognizing these mediating mechanisms opens opportunities for targeted interventions, such as enhancing SE or reducing nocturnal awakenings, with the potential to mitigate future sleep complaints and possibly slow cognitive decline [78,79]. In summary, the mediation analyses confirm that the objective–subjective sleep relationship is dynamic, with SE playing a central role, whereas TST and WASO exert indirect influences through their stability or decline over time, shaping the trajectory of subjective complaints [80].

## 5. Conclusions

The present study demonstrates that longitudinal trajectories of both subjective and objective sleep measures provide valuable insights into the differential progression of cognitive impairment in older adults. Among the groups, naMCI consistently exhibited the steepest decline in objective sleep quality, particularly in TST and SΕ and subjective sleep quality, highlighting this subgroup’s heightened vulnerability to sleep dysregulation. Insomnia symptoms and risk for sleep-disordered breathing also worsened over time, supporting the value of tracking sleep longitudinally rather than relying on single assessments [81]. Moreover, objective sleep parameters were closely linked to subjective complaints, reinforcing the interplay between actual sleep disruption and perceived sleep quality in progressive MCI. Overall, these findings highlight sleep as a dynamic behavioral marker of early cognitive decline and underscore the importance of routine longitudinal monitoring and early intervention, especially in naMCI populations.

## 6. Limitations

Although the present study employed a longitudinal design and incorporated both subjective and objective sleep assessments, some methodological limitations warrant consideration. While actigraphy serves as a practical and ecologically valid tool for monitoring sleep–wake patterns, it lacks the precision of polysomnography in capturing detailed sleep architecture, including alterations in REM and slow-wave sleep that may be critical in neurodegenerative trajectories. Despite these constraints, it proved to be a practical tool for our research objectives. Additionally, the lack of structural MRI and biomarkers, such as amyloid-β and PET scans, reduces the ability to link behavioral findings to neurobiological changes and prevents a clear connection between the variables of interest and biological alterations in individuals with MCI.

Furthermore, mediation analyses were performed on the total sample due to limited subgroup sizes. Exploratory checks suggested that effect patterns may differ across MCI subtypes; however, the high number of variables relative to subgroup sizes precluded separate analyses. This limitation has been explicitly noted to ensure transparency.

## 7. Future Implications—Clinical Use

The findings highlight the critical role of sleep assessment in the early identification of cognitive decline. Future research should aim to replicate these findings in larger, more diverse samples and incorporate more advanced tools such as polysomnography, to capture detailed sleep architecture (e.g., REM and slow-wave sleep) that is not measurable through actigraphy. The role of objective sleep in the perception of sleep quality offers a potential avenue for targeted interventions, suggesting that improving objective sleep parameters may enhance subjective sleep experience and possibly slow cognitive deterioration. Finally, integrating neuroimaging and biomarker data could further elucidate the mechanisms linking sleep disturbances and MCI progression. From a clinical perspective, SE, TST, and WASO emerge as the most promising objective indices, as they reliably predict future subjective complaints and differentiate MCI subtypes. Among subjective tools, PSQI and AIS may be particularly useful for routine screening, while STOP-BANG provides an accessible means of identifying comorbid sleep-disordered breathing risk. Together, these measures could form the basis of low-cost, scalable monitoring strategies for at-risk older adults.

## Figures and Tables

**Figure 1 diagnostics-15-02815-f001:**
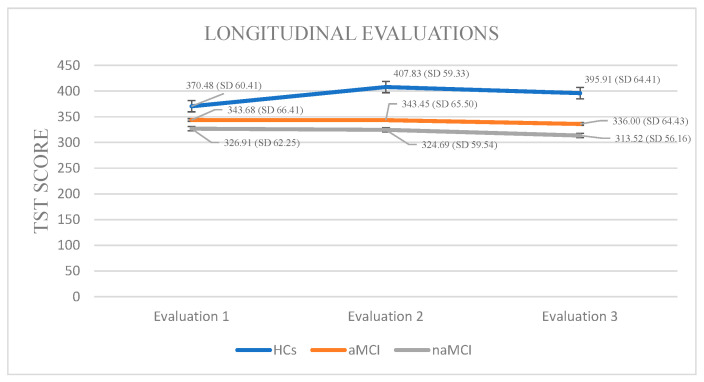
Performance of groups in TST in the three times of assessment.

**Figure 2 diagnostics-15-02815-f002:**
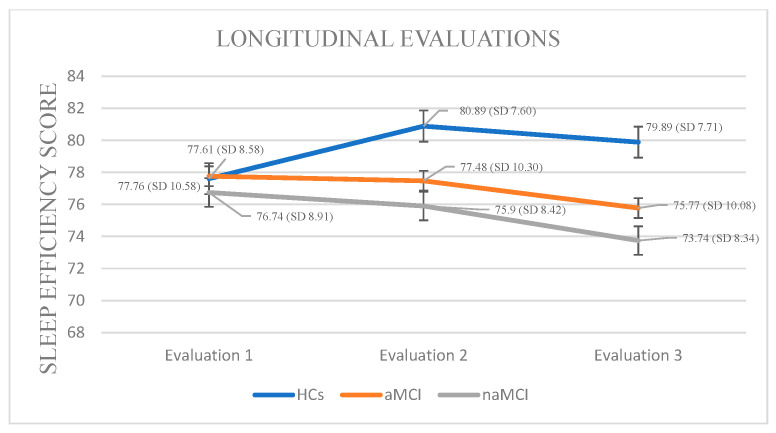
Performance of groups in SE in the three times of assessment.

**Figure 3 diagnostics-15-02815-f003:**
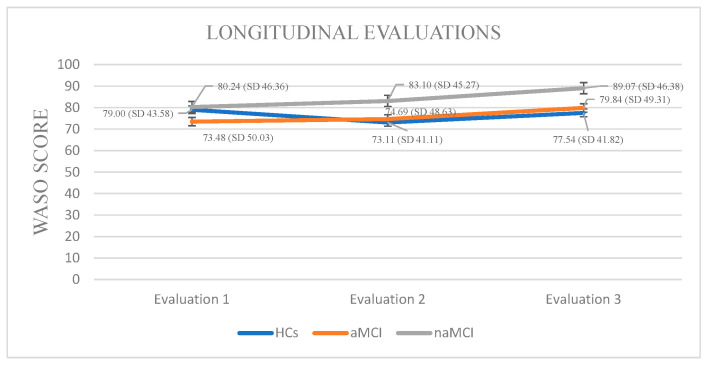
Performance of groups in WASO in the three times of assessment.

**Figure 4 diagnostics-15-02815-f004:**
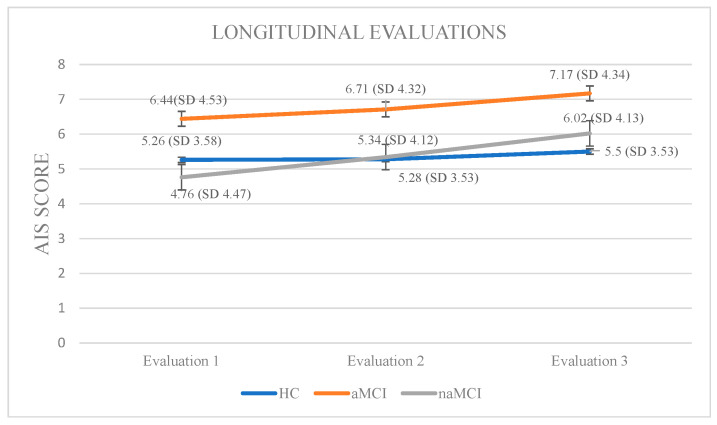
Performance of groups in AIS in the three times of assessment.

**Figure 5 diagnostics-15-02815-f005:**
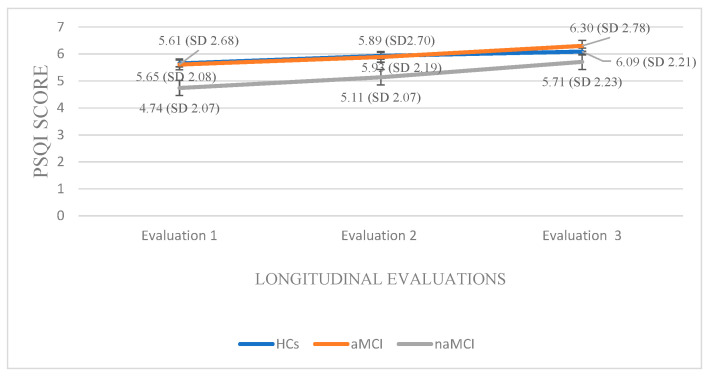
Performance of groups in PSQI in the three times of assessment.

**Figure 6 diagnostics-15-02815-f006:**
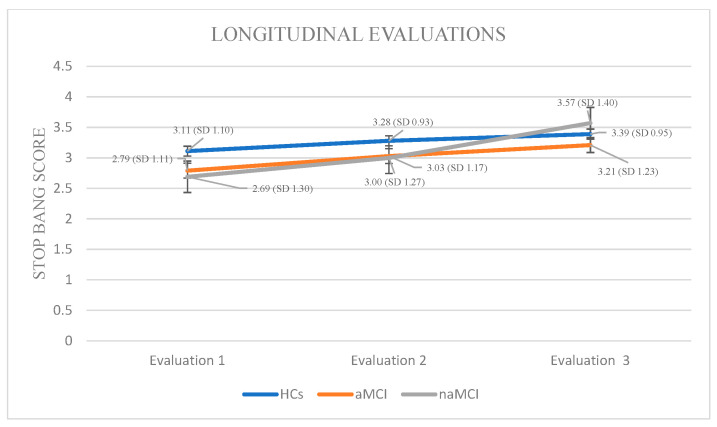
Performance of groups in STOP-BANG in the three times of assessment.

**Table 1 diagnostics-15-02815-t001:** Participants’ demographic characteristics.

	HC	aMCI	naMCI
	Mean (SD)	Mean (SD)	Mean (SD)
Age	70.28 (5.02)	71.09 (4.98)	69.07 (4.18)
Education	11.98 (3.09)	12.41 (3.18)	12.57 (3.39)
Gender (f/m)	36/10	51/24	39/19

**Table 2 diagnostics-15-02815-t002:** The complex role of objective sleep assessments in the formulation and change of subjective sleep estimations longitudinally.

Pathway	Estimate	Std. Error	z-Value	*p*	95% CI Lower	95% CI Upper
Direct effects						
SE (2nd assessment) → AIS (3rd assessment)	0.084	0.036	2.333	0.020	0.013	0.155
TST (2nd assessment) → AIS (3rd assessment)	−0.003	0.003	−0.927	0.354	−0.010	0.003
WASO (2nd assessment) → AIS (3rd assessment)	−0.003	0.007	−0.431	0.666	−0.016	0.010
TST (2nd assessment) → PSQI (3rd assessment)	−0.001	0.003	−0.344	0.731	−0.007	0.005
WASO (2nd assessment) → PSQI (3rd assessment)	0.008	0.006	1.401	0.161	−0.003	0.020
SE (2nd assessment) → PSQI (3rd assessment)	0.047	0.032	1.501	0.133	−0.014	0.109
TST (2nd assessment) → STOP-BANG (3rd assessment)	0.003	0.003	1.183	0.237	−0.002	0.008
WASO (2nd assessment) → STOP-BANG (3rd assessment)	0.001	0.005	0.213	0.832	−0.009	0.012
SE (2nd assessment) → STOP-BANG (3rd assessment)	0.005	0.029	0.161	0.878	−0.051	0.060
Indirect effects						
TST (2nd assessment) → AIS (2nd assessment) → AIS (3rd assessment)	−0.011	0.005	−2.098	0.036	−0.021	−7.060 × 10^−4^
SE (2nd assessment) → SE (3rd assessment) → AIS (3rd assessment)	−0.083	0.035	−2.315	0.021	−0.151	−0.013
WASO (2nd assessment) → PSQI (2nd assessment) → PSQI (3rd assessment)	−0.016	0.007	−2.217	0.027	−0.030	−0.002
SE (2nd assessment) → PSQI (2nd assessment) → PSQI (3rd assessment)	−0.102	0.039	−2.593	0.010	−0.178	−0.025

## Data Availability

The data presented in this study are available on request from the corresponding author, as they are part of my doctoral dissertation, which has not yet been fully published.

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
