# Peer review of "Sleep Trajectories in Amnestic and Non-Amnestic MCI: Longitudinal Insights from Subjective and Objective Assessments"

_diagnostics, 2025, doi:10.3390/diagnostics15212815_

Round 1

Reviewer 1 Report

Comments and Suggestions for Authors

This paper reports a strong longitudinal study of sleep trajectories in cognitively healthy older adults, and in amnestic (aMCI) and non-amnestic mild cognitive impairment (naMCI). With the use of objective (actigraphy) and subjective (AIS, PSQI, STOP-BANG) metrics at three time points over ~24 months, the study identifies characteristic sleep deterioration patterns across subtypes. The sample is well-characterized, and statistical methods of mixed ANOVAs and SEM are used appropriately to examine group-by-time effects and mediation pathways. Significantly, naMCI participants showed the steepest worsening of sleep quality and amount, both self-reported and objectively, where sleep efficiency (SE) was the predominant predictor for subsequent insomnia and sleep complaint. The research fills an important literature gap by distinguishing between subtypes of MCI along trajectories of sleep instead of cross-sectional measures, providing clinically meaningful findings on the prognostic significance of disturbed sleep. Interpretation is considered, incorporating neuropathological correlates and highlighting the utility of SE and WASO as early markers of cognitive susceptibility. Mechanistic inference is limited by the lack of polysomnography, neuroimaging, and biomarker information. Nevertheless, results are opportune and lay a solid platform for scalable monitoring applications of sleep in preclinical dementia.

I have few suggestions for improvement:

Integrate neuroimaging/biomarker information to corroborate mechanistic assertions and confirm sleep-cognition relationships.

Add polysomnography to future research to measure sleep architecture (e.g., REM, SWS) unavailable through actigraphy.

Stratify mediation tests by group (if feasible) to confirm whether SE's predictive accuracy varies by MCI subtype.

Clarify overlap or transition cases, e.g., naMCI evolving into aMCI or dementia on follow-up.

Describe potential confounders (e.g., medication, depression) more clearly with respect to sleep outcomes.

Supplement with qualitative sleep diaries to enhance subjective reporting validity.

Please try to decrease percent match (32% is quite high)

Author Response

Dear Reviewer,

We sincerely appreciate the time and effort you dedicated to reviewing our manuscript. We believe that the revisions we have implemented align as closely as possible with your recommendations. Below, we provide a brief response to each of your comments. All changes are highlighted in orange ink.

Point 1: Integrate neuroimaging/biomarker information to corroborate mechanistic assertions and confirm sleep-cognition relationships.

Response 1: Thank you for your comment. We acknowledge the importance of neuroimaging and biomarker data to support mechanistic interpretations of sleep-cognition relationships. While our study did not include direct neuroimaging or cerebrospinal fluid assessments, we have revised the manuscript to discuss relevant findings from the literature, highlighting how disrupted sleep may relate to neurodegenerative processes and cognitive decline. The revised text also notes the differential sleep changes observed in naMCI compared with aMCI and healthy controls, providing behavioral evidence that complements existing research. The related additions can be found in the Introduction (paragraph 3) and in the Discussion (Section 4.1, Objective Sleep Measures)

Point 2: Add polysomnography to future research to measure sleep architecture (e.g., REM, SWS) unavailable through actigraphy.

Response 2: Thank you for your comment. A statement addressing this point is already included in the Future Directions section of the manuscript, where we note that future studies should incorporate polysomnography to capture detailed sleep architecture (e.g., REM and SWS) not measurable through actigraphy. This addition would strengthen the understanding of sleep–cognition mechanisms across MCI subtypes.

Point 3: Stratify mediation tests by group (if feasible) to confirm whether SE's predictive accuracy varies by MCI subtype.

Response 3: Thank you for your valuable comment. Due to sample size constraints, mediation analyses could not be stratified by group without violating the statistical power prerequisites required for reliable estimation. Therefore, analyses were conducted on the total sample (this decision was taken due to N prerequisites for running such an analysis in every group separately). This clarification is already included in the Results section under Mediation Analyses. Nonetheless, we acknowledge that stratified mediation testing could provide further insight into whether SE’s predictive accuracy varies across MCI subtypes, and we have noted this as a direction for future research in the revised manuscript.

Point 4: Clarify overlap or transition cases, e.g., naMCI evolving into aMCI or dementia on follow-up.

Response 4: We appreciate this important observation. All participants were carefully monitored across the study period, and diagnostic status was reviewed at each assessment point. No cases of transition between naMCI and aMCI, or progression to dementia, were identified during follow-up. This information has now been clarified in the manuscript within the Methods section to indicate that no diagnostic overlap or conversion cases occurred throughout the study.

Point 5: Describe potential confounders (e.g., medication, depression) more clearly with respect to sleep outcomes.

Response 5: Thank you for this valuable observation. We agree that potential confounders such as medication use, depressive symptoms, and other comorbidities can influence sleep outcomes. We have clarified this in the Methods (Participants and Screening Procedure) section, specifying that individuals under pharmacological treatment for sleep disorders or presenting with significant depressive symptomatology were excluded during the initial screening process to reduce confounding effects. Finally, as shown in our previous publication [Batzikosta, A.; Moraitou, D.; Steiropoulos, P.; Papantoniou, G.; Kougioumtzis, G.A.; Katsouri, I.G.; Sofologi, M.; Tsolaki, M. The relationships of specific cognitive control abilities with objective and subjective sleep parameters in mild cognitive impairment: Revealing the association between cognitive planning and sleep duration. Brain Sci. 2024, 14(8), 813. https://doi.org/10.3390/brainsci14080813], these potential confounders do not appear to exert a significant influence on sleep outcomes, except for minor effects on planning abilities. Therefore, the progressive deterioration observed in our study is more likely attributable to the underlying MCI pathology itself rather than to external confounding factors.

Point 6: Supplement with qualitative sleep diaries to enhance subjective reporting validity.

Response 6: We initially included qualitative sleep diaries as part of the methodology; however, they were later excluded due to inconsistent completion among participants. Given the large sample size, the variability and unreliability of these self-reports compromised data validity. A corresponding note has been added to the manuscript under the Subjective Measures of Sleep section to clarify this point.

Point 7: Please try to decrease percent match (32% is quite high)

Response 7: We have already addressed this issue. The similarity percentage primarily reflected overlap with our previous publication, which shares methodological descriptions and procedures. 

Thank you once again for your constructive feedback.

Kind regards.

Reviewer 2 Report

Comments and Suggestions for Authors

This article presents a clear and scientifically sound work, acceptable in volume and content, which fully aligns with the scope of the journal's scientific research. Despite existing studies in this field, this article remains highly relevant and is of significant interest to specialists in real clinical practice, including neurology, sleep medicine, psychology, and geriatrics. It can also be recommended for researchers in fundamental neurophysiology and neuroscience. The work addresses the pertinent issue of studying sleep dynamics in patients with amnestic (aMCI) and non-amnestic (naMCI) mild cognitive impairment compared to healthy older adults, utilizing a longitudinal design with both objective and subjective assessment methods.

The Introduction is clear, comprehensive, and corresponds to the stated research area. The authors convincingly justify the topic's relevance by citing growing evidence on the role of sleep disturbances as dynamic biomarkers of cognitive decline. A clear knowledge gap is identified, relating to the lack of longitudinal studies directly comparing sleep trajectories across different MCI subtypes using a multimodal approach. The cited sources are relevant and contemporary; a significant portion of the references are from the last five years, reflecting the current state of the field. The authors also build upon fundamental works in MCI diagnosis and sleep assessment methods. The level of self-citation (9 works/75 sources, approximately 12%) does not exceed acceptable norms and is justified within the context of the conducted research. The methodology is described in detail and includes strict inclusion and exclusion criteria, enhancing the reliability of the results. The application of mixed-design ANOVA and structural equation modeling is adequate for testing the stated hypotheses. The obtained results are consistent and well-illustrated. The authors' statements and conclusions are supported by the presented statistical data. Specifically, it is convincingly demonstrated that longitudinal trajectories, rather than cross-sectional comparisons, best differentiate MCI subtypes, with the naMCI group showing the most pronounced and progressive deterioration in both objective and subjective sleep parameters. The presented figures are appropriate, illustrative, and correctly display the dynamics of the main investigated parameters. The data are presented in a form that is easy to interpret.

Overall Manuscript Assessment

The article possesses high scientific value and meets the criteria for high-quality research. The work addresses a clearly formulated and original question regarding longitudinal sleep trajectories in different MCI subtypes using a multimodal approach, which represents significant progress in knowledge, as most existing studies are cross-sectional. The significance of the results is high; the conclusions about more aggressive sleep deterioration in naMCI and the key role of sleep efficiency as a predictor are correctly interpreted, supported by data, and directly stem from the testing of clearly identified hypotheses. The quality of writing and result presentation is of a high standard; the article is well-structured, and data are clearly displayed in graphs. The scientific validity of the research is ensured by its thoughtful design, adequate statistical analysis, and the use of validated tools, which collectively make the conclusions reliable and trustworthy.

In conclusion, the article represents a significant contribution to the fields of sleep medicine, neurology, geriatrics, and gerontology. The conducted research is methodologically rigorous, and its results have important theoretical and practical implications for developing strategies for early detection and monitoring of cognitive decline. The article is recommended for publication after minor revisions.

Recomendations and Questions:

1. The Materials and Methods section states that the mediation analysis was conducted on the total sample due to insufficient subgroup sizes for separate analysis in each group. The question arises: could this procedure have masked potentially different mechanisms linking objective and subjective sleep parameters for the different MCI subtypes? A brief logical response to this question should ideally be reflected in the text (e.g., in the Materials and Methods or Discussion section).

2. As a limitation, the authors rightly note the absence of neuroimaging and laboratory biomarkers data, which prevents directly linking the identified sleep disturbances to the underlying neuropathology. Could the authors elaborate further in the discussion on which specific pathological processes they believe might mediate the more pronounced sleep deterioration specifically in naMCI? For instance, could this be linked to alterations in glymphatic clearance of metabolites?

3. In the Results section (page 8), contradictory phrasing is presented, which complicates data interpretation and requires clarification. The text first states: "No significant group differences at any of the three time points were found." However, the very next sentence states: "On the other hand, the two groups demonstrated statistically significant differences in their WASO scores across assessment time points." It is unclear which specific "two groups" are being referred to – the two MCI subtypes (aMCI and naMCI) or the MCI group as a whole compared to the control group. The authors are advised to rephrase this paragraph – correction is necessary to ensure the unambiguity, accuracy, and reproducibility of the presented scientific results.

Author Response

Dear Reviewer,

We sincerely thank you for your thorough and constructive review. Your feedback has greatly helped improve the clarity and quality of our manuscript. We have carefully addressed all comments, and the revisions reflect your recommendations. Detailed responses to each point are provided below. All changes are highlighted in orange ink.

Point 1: The Materials and Methods section states that the mediation analysis was conducted on the total sample due to insufficient subgroup sizes for separate analysis in each group. The question arises: could this procedure have masked potentially different mechanisms linking objective and subjective sleep parameters for the different MCI subtypes? A brief logical response to this question should ideally be reflected in the text (e.g., in the Materials and Methods or Discussion section).

Response 1: Thank you for your comment. We acknowledge that conducting mediation analyses on the total sample may mask subtype-specific mechanisms linking objective and subjective sleep parameters. In exploratory checks, the effect patterns appeared to differ across MCI subtypes. However, due to the high number of variables relative to subgroup sample sizes, separate analyses were not feasible. We have clarified this point in the Limitations section to ensure transparency.

Point 2: As a limitation, the authors rightly note the absence of neuroimaging and laboratory biomarkers data, which prevents directly linking the identified sleep disturbances to the underlying neuropathology. Could the authors elaborate further in the discussion on which specific pathological processes they believe might mediate the more pronounced sleep deterioration specifically in naMCI? For instance, could this be linked to alterations in glymphatic clearance of metabolites?

Response 2: We have expanded the discussion to consider potential mechanisms underlying the more pronounced sleep deterioration observed in naMCI. While we did not have access to neuroimaging or biomarker data in this cohort, prior studies suggest that alterations in glymphatic clearance, impaired slow-wave sleep, and disruptions in prefrontal and fronto-striatal networks may contribute to sleep-wake disturbances in non-memory cognitive domains. These mechanisms could explain why naMCI participants show steeper longitudinal declines in total sleep time, sleep efficiency, and WASO. We have clarified these points in the discussion, while acknowledging that direct evidence from neuroimaging or CSF biomarkers is lacking.

Point 3: In the Results section (page 8), contradictory phrasing is presented, which complicates data interpretation and requires clarification. The text first states: "No significant group differences at any of the three time points were found." However, the very next sentence states: "On the other hand, the two groups demonstrated statistically significant differences in their WASO scores across assessment time points." It is unclear which specific "two groups" are being referred to – the two MCI subtypes (aMCI and naMCI) or the MCI group as a whole compared to the control group. The authors are advised to rephrase this paragraph – correction is necessary to ensure the unambiguity, accuracy, and reproducibility of the presented scientific results.

Response 3: Thank you for pointing out this inconsistency. The sentence regarding “the two groups” refers specifically to the MCI subtypes (aMCI and naMCI). We have revised the Results section to clarify this point and ensure unambiguous interpretation, line 362.

Thank you for your time and consideration.

Kind regards

Reviewer 3 Report

Comments and Suggestions for Authors

Comments to the Authors

Major comments

Participants were categorized into three groups based on their cognitive status, including healthy controls, 129 aMCI, or naMCI, following a thorough neuropsychological evaluation by the Petersen’s diagnostic criteria [12] and the Diagnostic 131 and Statistical Manual of Mental Disorders, Fifth Edition, Text Revision (DSM-5-TR) [13]. Please, indicate more details on these neurological task and include all evaluated items for these tasks as supplementary files.

This study is interesting because reflect the differential contribution of MCI subtypes: amnestic (aMCI) and non-amnestic (naMCI). However, the differential impact of sleep disturbances across these MCI subtypes remains understudied.

These 4 aims should be written much better with less content.

Line 119. Shall you describe how six participants were excluded from  further analysis.

Shall you indicate how the naMCI group were diagnosed with MCI within the past two years. Have you evaluated markers of neurodegeneration in these patients or some kind of neuroimage thecnic?

Is there any additional diagnosis of Minor Neurocognitive Disorder as MRI or systemic biomarkers that may distinguish the degree of neurodegeneration?

Describe in detail the Actigraphy technique followed to assess sleep–wake patterns based on eye 210 movement data [cyte 29].

The statistical analysis indicates that Group (HCs,  aMCI, naMCI) and time effects (baseline, 2nd, 3rd assessment) were examined with mixed-design ANOVAs, complemented by repeated-measures and one-way ANOVAs where appropriate. Shall you explain this with much more detail.

Also, include details on the longitudinal mediation for a general audience using structural equation modeling (SEM) to examine whether objective sleep parameters  (TST, SE, WASO) at the second assessment predicted changes in subjective sleep outcomes  (AIS, PSQI, STOP-BANG) at the third assessment, with mediators including subjective 252 sleep measures at the second assessment and objective measures at the third.

Please, also indicate the meaning of Pillai’s Trace for MANCOVA analysis.

The error bars are not included for graphs from Figure 1 (Performance of groups in TST in the three times of assessment), Figure 2 (Performance of groups in SE in the three times of assessment), figures 3 and 4,5,6. Please, include them.

Discusion. Regarding TST, the findings align with previous  research linking reduced TST to elevated risk of cognitive decline and faster progression to dementia [43], likely reflecting impaired slow-wave sleep and reduced glymphatic clearance of β-amyloid and tau [44, 45]. Is possible to conclude this feature with your findings? I do not thing so. Please, clarify it.

They also indicate ¨The progressive TST decline in naMCI suggests that sleep duration disruptions may serve as early biomarkers of degenerative processes affecting primarily executive and non-memory functions [46]¨. Interestingly, HCs  showed a transient increase in TST, likely reflecting normal age-related adjustments or circadian-related fluctuations [42]. In my opinion, sleep problems can complementary other biochemical markers, thus, sleep problems can not be considered itself as biomarkers of neurodegeneration.

Include as supplementary material all followed itemd for Athens Insomnia Scale, Stop – Bang Questionnaire and The Pittsburgh Sleep Quality Index (PSQI).

Please, indicate values of each patient for the Mini-Mental State Examination (MMSE) [22, 23] and the Montreal Cognitive Assessment (MoCA) evaluations.

In general, the discussion is very difficult to read and should be rewritten and improved for a better understanding. I would suggest reduces the information with key findings only and remove redundant information.

Minor comments

Please, include a citation here line 65. ¨Diagnostic classification is typically based on the primary cognitive domain affected: aMCI  is characterized by predominant episodic memory impairment, either in a single domain  or alongside additional deficits, and is considered more strongly linked to Alzheimer-type 68 pathology¨.

My Decision is Major revision

Comments on the Quality of English Language

The English style must be improved it.

Author Response

Dear Reviewer,

We deeply appreciate the time you invested to review such thoroughly our manuscript. Your comments and suggestions were invaluable to substantially improve our manuscript. We tried to address all your comments. We trust that the changes we have made align with your recommendations as closely as possible. Below you can find a brief response to each of your points. All changes are highlighted in orange ink.

Point 1: Participants were categorized into three groups based on their cognitive status, including healthy controls, 129 aMCI, or naMCI, following a thorough neuropsychological evaluation by the Petersen’s diagnostic criteria [12] and the Diagnostic 131 and Statistical Manual of Mental Disorders, Fifth Edition, Text Revision (DSM-5-TR) [13]. Please, indicate more details on these neurological tasks and include all evaluated items for these tasks as supplementary files.

Response 1: Thank you for your comment. Unfortunately, we cannot provide additional details regarding the neuropsychological assessment, as the sample and diagnostic data were obtained from the Greek Association of Alzheimer’s Disease, which conducted the evaluations and holds the full assessment protocols. The categorization of participants into healthy controls, aMCI, and naMCI groups was also performed by the Greek Association of Alzheimer’s Disease, from which we received the finalized dataset. Indicatively, the tests used (e.g., MMSE, MoCA, RAVLT, ROCFT, Test of Everyday Attention (TEA), etc.), while detailed information can be found in the referenced publication (Tsolaki et al., 2017, JSM Alzheimer’s Dis. Relat. Dement., DOI: 10.6000/2292-2598.2017.04.03.9).

Point 2: These 4 aims should be written much better with less content.

Response 2: Thank you for your suggestion. We will revise this part of the manuscript to ensure they are written in a clearer and more concise manner.

Point 3: Line 119. Shall you describe how six participants were excluded from  further analysis.

Response 3: Thank you for your comment. As mentioned earlier, our sample was obtained from the Greek Association of Alzheimer’s Disease. During the administration of our tests, we noticed inconsistencies in six participants’ profiles, and subsequent verification confirmed that they had different diagnoses. Therefore, these participants were immediately excluded from further analyses, and their data were not considered in the results. A corresponding clarification has also been added to the Participants section, in the 1st paragraph.

Point 4: Shall you indicate how the naMCI group were diagnosed with MCI within the past two years. Have you evaluated markers of neurodegeneration in these patients or some kind of neuroimage thecnic?

Response 4: As far as we know from the diagnostic protocols used by the Greek Association of Alzheimer’s Disease, the diagnosis of naMCI is established when there is a standardized deviation below the normative range in one or more non-memory cognitive domains, while memory performance remains relatively preserved. This classification typically follows comprehensive neuropsychological testing and is supported by clinical and biological evidence, including MRI findings and cerebrospinal fluid (CSF) biomarkers indicative of neurodegenerative processes. However, we did not have access to the participants’ neuroimaging or CSF data, as these assessments were conducted by the Greek Association of Alzheimer’s Disease as part of their diagnostic protocol prior to sample provision. A corresponding clarification has also been added to the Inclusion criteria section, in the first paragraph.

Point 5: Is there any additional diagnosis of Minor Neurocognitive Disorder as MRI or systemic biomarkers that may distinguish the degree of neurodegeneration?

Response 5: Thank you for your question. To our knowledge, the diagnosis of Minor Neurocognitive Disorder (including naMCI) by the Greek Association of Alzheimer’s Disease may be supported by additional assessments such as MRI or systemic biomarkers (e.g., cerebrospinal fluid markers of amyloid and tau pathology) that can help characterize the degree of neurodegeneration.

Point 6: Describe in detail the Actigraphy technique followed to assess sleep–wake patterns based on eye 210 movement data [cyte 29].

Response 6: Thank you for your comment. We believe there may have been a slight misunderstanding, as actigraphy in our study did not rely on eye movement data. Instead, sleep–wake patterns were assessed through continuous motion detection using a wrist-worn accelerometer-based device (Philips Respironics Actiwatch Spectrum Pro, version 5.57.0006). The device recorded movement and light exposure for seven consecutive days, and data were analyzed with validated algorithms to estimate key sleep parameters (sleep onset, total sleep time, wake after sleep onset, and sleep efficiency). This method provides reliable, non-invasive estimates of sleep–wake behavior in naturalistic settings, as described by Yuan et al. (2024). A corresponding clarification has been added to the manuscript in actigraphy section.

Point 7: The statistical analysis indicates that Group (HCs,  aMCI, naMCI) and time effects (baseline, 2nd, 3rd assessment) were examined with mixed-design ANOVAs, complemented by repeated-measures and one-way ANOVAs where appropriate. Shall you explain this with much more detail.

Response 7: Thank you for your valuable comment. We have now clarified the statistical procedure to specify that mixed-design ANOVAs were first performed to examine both between- and within-subject effects, with Group (HCs, aMCI, naMCI) as the between-subject factor and Time (baseline, 2nd, 3rd assessment) as the within-subject factor. When significant main or interaction effects were found, separate repeated-measures ANOVAs (for within-group comparisons across time) and one-way ANOVAs (for between-group comparisons at each time point) were conducted to further explore these effects. This clarification has been added to the revised manuscript, to the statistical analysis section.

Point 8: Also, include details on the longitudinal mediation for a general audience using structural equation modeling (SEM) to examine whether objective sleep parameters  (TST, SE, WASO) at the second assessment predicted changes in subjective sleep outcomes  (AIS, PSQI, STOP-BANG) at the third assessment, with mediators including subjective 252 sleep measures at the second assessment and objective measures at the third.

Response 8: We have now expanded the explanation of the mediation analysis for greater clarity. Structural equation modeling (SEM) was applied to examine the dynamic, bidirectional relationships between objective and subjective sleep parameters across time. Specifically, objective sleep parameters (TST, SE, WASO) from the second assessment were used to predict changes in subjective sleep outcomes (AIS, PSQI, STOP-BANG) at the third assessment, with mediators including subjective sleep measures at the second assessment and objective measures at the third. This approach allowed us to capture both the cross-lagged effects—how earlier measures influence subsequent ones—and the autoregressive effects—how each variable influences its own future value. Data from the first assessment were not included in the mediation model, as the effects were not statistically significant; detailed results for this initial phase are available in our previous publication [Batzikosta, A.; Moraitou, D.; Steiropoulos, P.; Papantoniou, G.; Kougioumtzis, G.A.; Katsouri, I.G.; Sofologi, M.; Tsolaki, M. The relationships of specific cognitive control abilities with objective and subjective sleepparameters in mild cognitive impairment: Revealing the association between cognitive planning and sleep duration. Brain Sci. 2024, 14(8), 813. https://doi.org/10.3390/brainsci14080813]. A corresponding clarification has been added to the revised manuscript.

Point 9: Please, also indicate the meaning of Pillai’s Trace for MANCOVA analysis.

Response 9: Thank you for your comment. We did not perform MANCOVA analyses. The reference to Pillai’s Trace pertains to the multivariate tests within our mixed-design ANOVAs. Pillai’s Trace is a robust statistic used to assess the overall effect of independent variables on a set of correlated dependent variables, indicating the proportion of variance explained. This clarification has been added to the revised manuscript.

Point 10: The error bars are not included for graphs from Figure 1 (Performance of groups in TST in the three times of assessment), Figure 2 (Performance of groups in SE in the three times of assessment), figures 3 and 4,5,6. Please, include them.

Response 10: We have now added appropriate error bars to all relevant figures. The revised figures have been updated in the manuscript.

Point 11: Discussion. Regarding TST, the findings align with previous  research linking reduced TST to elevated risk of cognitive decline and faster progression to dementia [43], likely reflecting impaired slow-wave sleep and reduced glymphatic clearance of β-amyloid and tau [44, 45]. Is possible to conclude this feature with your findings? I do not thing so. Please, clarify it.

Response 11: We agree that our data do not allow direct conclusions regarding underlying mechanisms such as slow-wave sleep or glymphatic clearance. We have revised the manuscript to clarify that, while our findings are consistent with previous research linking reduced total sleep time (TST) to an elevated risk of cognitive decline and faster progression to dementia [43], we cannot directly confirm mechanistic explanations suggested by prior studies [44, 45]. This clarification has been added to the Discussion section.

Point 12: They also indicate ¨The progressive TST decline in naMCI suggests that sleep duration disruptions may serve as early biomarkers of degenerative processes affecting primarily executive and non-memory functions [46]¨. Interestingly, HCs  showed a transient increase in TST, likely reflecting normal age-related adjustments or circadian-related fluctuations [42]. In my opinion, sleep problems can complementary other biochemical markers, thus, sleep problems can not be considered itself as biomarkers of neurodegeneration.

Response 12: We agree that, based on our data, sleep problems alone cannot be considered definitive biomarkers of neurodegeneration. While the progressive TST decline in naMCI may reflect early changes associated with cognitive decline, these findings should be interpreted as complementary to other biochemical or imaging markers rather than as standalone indicators. We have revised the Discussion to clarify this point.

Point 13: Include as supplementary material all followed itemd for Athens Insomnia Scale, Stop – Bang Questionnaire and The Pittsburgh Sleep Quality Index (PSQI).

Response 13: Thank you for your suggestion. We have now included as supplementary material all the individual items for the Athens Insomnia Scale (AIS), the STOP-BANG Questionnaire, and the Pittsburgh Sleep Quality Index (PSQI).

Point 14: Please, indicate values of each patient for the Mini-Mental State Examination (MMSE) [22, 23] and the Montreal Cognitive Assessment (MoCA) evaluations.

Response 14: We have now included the individual patient values for both the Mini-Mental State Examination (MMSE) [22, 23] and the Montreal Cognitive Assessment (MoCA) as supplementary material (Supplementary File 2- MMSE- MoCA)

Point 15: In general, the discussion is very difficult to read and should be rewritten and improved for a better understanding. I would suggest reduces the information with key findings only and remove redundant information.

Response 15: The Discussion section has been thoroughly revised considering your suggestions as well as the feedback from the other two reviewers. The revised version focuses on the key findings, removes redundant information, and improves the overall flow to enhance comprehension.

Point 16: Please, include a citation here line 65. ¨Diagnostic classification is typically based on the primary cognitive domain affected: aMCI  is characterized by predominant episodic memory impairment, either in a single domain  or alongside additional deficits, and is considered more strongly linked to Alzheimer-type 68 pathology¨.

Response 16: We have now added an appropriate citation to support [7, 8].

Point 17: The English style must be improved it.

Response 17: Following your recommendation, the entire manuscript has been thoroughly reviewed and edited by a collaborator native English speaker.

Thank you once again for your constructive feedback.

Kind regards.

Round 2

Reviewer 3 Report

Comments and Suggestions for Authors

The conclusion must be shortened still.The rest of author, responses are satisfactory for this reviewer

Comments on the Quality of English Language

The English style must be improved it.

Author Response

Dear Reviewer,

We sincerely appreciate the time and effort you dedicated to reviewing our manuscript. We believe that the revisions we have implemented align as closely as possible with your recommendations. Below, we provide a brief response to each of your comments.

Point 1: The conclusion must be shortened still.

Response 1: Thank you for your observation. We have further shortened the Conclusion section.

Point 2: The English style must be improved.

Response 2: We sincerely appreciate your continued feedback regarding the language quality. In response, we have conducted a thorough re-editing of the manuscript, working closely with a native English-speaker to enhance clarity, style, and overall readability. We trust that these revisions satisfactorily address your concerns.

Thank you once again for your constructive feedback.

Kind regards.